# Multi-omic data integration enables discovery of hidden biological regularities

Ali Ebrahim[1,*], Elizabeth Brunk[1,2,*], Justin Tan[1,*], Edward J. O'Brien[3], Donghyuk Kim[1], Richard Szubin[1], Joshua A. Lerman[3], Anna Lechner[4], Anand Sastry[1], Aarash Bordbar[1], Adam M. Feist[1,2] & Bernhard O. Palsson[1,2,3,5]

Rapid growth in size and complexity of biological data sets has led to the 'Big Data to Knowledge' challenge. We develop advanced data integration methods for multi-level analysis of genomic, transcriptomic, ribosomal profiling, proteomic and fluxomic data. First, we show that pairwise integration of primary omics data reveals regularities that tie cellular processes together in *Escherichia coli*: the number of protein molecules made per mRNA transcript and the number of ribosomes required per translated protein molecule. Second, we show that genome-scale models, based on genomic and bibliomic data, enable quantitative synchronization of disparate data types. Integrating omics data with models enabled the discovery of two novel regularities: condition invariant *in vivo* turnover rates of enzymes and the correlation of protein structural motifs and translational pausing. These regularities can be formally represented in a computable format allowing for coherent interpretation and prediction of fitness and selection that underlies cellular physiology.

[1] Department of Bioengineering, University of California, San Diego, 9500 Gilman Drive, Mail Code 0412, La Jolla, California 92093, USA. [2] The Novo Nordisk Foundation Center for Biosustainability, Technical University of Denmark, Kemitorvet, Building 220 DK-2800 Kongens Lyngby, Denmark. [3] Bioinformatics and Systems Biology Program, University of California, San Diego, California 92093, USA. [4] Department of Chemical and Biomolecular Engineering, University of California, Berkeley, California 94720, USA. [5] Department of Pediatrics, University of California, San Diego, California 92093, USA. * These authors contributed equally to this work. Correspondence and requests for materials should be addressed to B.O.P. (email: palsson@ucsd.edu).

Progress of the biological sciences in the era of big data will depend on how we address the following question: 'How do we connect multiple disparate data types[1] to obtain a meaningful understanding of the biological functions of an organism[2]?' Owing to large-scale improvements in omics technologies, we can now quantitatively track changes in biological processes in unprecedented detail[3,4]. Although such measurements span a diverse range of cellular activities, developing an understanding of how these data types quantitatively relate to one another and to the phenotypic characteristics of the organism remains elusive. This issue is central to the so-called Big Data to Knowledge (BD2K) grand challenge, which aims to integrate multiple disparate data types into a biologically meaningful, multi-level structure[1,2].

Interpretation of disparate data requires understanding how the primary measurements of different omics data are quantitatively coupled to one another[5]. We approach this task by identifying regularities (relationships between biological data types that remain relatively constant across conditions) between pairwise omics data types. Although some regularities can readily be discovered through direct pairwise omics data comparisons, we find that other regularities emerge only through more intricate analysis leveraged by mechanistically based network reconstructions[6]. Such reconstructions can be used as a context for poly-omic data integration and analysis[6,7], and, when combined with constraint-based modelling approaches[8,9], provide important links between omics data and phenotypic characteristics of the organism.

As we will show, this approach leads to a comprehensive synchronization of poly-omic data with computed growth states. The approach directly addresses the BD2K grand challenge and is made conceptually accessible by tracing the 'information flow' through the familiar 'central dogma', to establish relationships between measurements and cell physiology (Fig. 1).

## Results

**Pairwise ratios of data types are highly correlated.** First, we examine the information flow from transcription to translation, to protein production, by identifying correlations across primary omics data types, such as RNAseq[10], ribosome profiling[11–13] and proteomics[14], collected for *Escherichia coli* batch growth on glucose, fumarate, pyruvate and acetate (Fig. 1, 'primary data box'). We found relatively poor correlations of messenger RNA to protein across conditions ($r^2 < 0.4$), consistent with previous studies[15,16]. Stronger correlations ($r^2 > 0.8$) emerge when analysing the ratio of protein per mRNA ($\rho_{PM}$) on a per-gene basis (the difference between peptide abundance and relative mRNA read counts per gene for multiple growth conditions; Supplementary Fig. 1a). Computing the median coefficient of variation shows that changes in $\rho_{PM}$ across conditions are relatively invariant. In addition, we find the number of ribosomes required (ribosome occupancy of mRNA) per protein translated is also relatively invariant across all four conditions ($r^2 > 0.7$; Supplementary Fig. 1b).

**Translation rate is linked to protein secondary structure.** Second, we examined pairwise relationships between other omics data types, such as ribosome profiling, proteomics and fluxomics, by integrating these data types into next-generation genome-scale models (Fig. 1, 'integration with genome-scale models of metabolism (GEMs) box'). GEMs are based on the annotated sequence and analysis of the bibliome for functionally annotated gene products[6]. The most recent generations of genome-scale models incorporate protein structural information[17,18] and allow

for the computation of the synthesis of the entire proteome of a cell in addition to the balanced use of its metabolic network[19]. These models can integrate multiple layers of biological organization to balance the use of all cellular components, to achieve a cellular state. It can thus extend our understanding of how information flows from translation to protein folding and catalysis, and its role in producing whole-cell functions.

We examined how information flows during protein translation, which includes protein folding. Recent studies indicate a possible link between translation speed and proper folding[12,20]. Analysis of translational pausing has typically been approached from a sequence-based viewpoint[20]. Here we approach this analysis from a different perspective, by correlating the occurrence of translational pausing on a transcript to the location of nearby protein secondary (2°) structure motifs (Fig. 2). The establishment of this correlation is based on the following: (1) ribosome profiling[11–13], which provides ample information on the queuing of ribosomes along mRNA transcripts; and (2) a recent network reconstruction that contains comprehensive protein structural information linked to the translated protein at the proteome scale[18].

Several striking regularities in translational pausing and protein structure are consistently observed across multiple growth conditions in *E. coli*, which suggest the co-translational folding of intermediate secondary structure motifs inside the ribosome exit tunnel (Fig. 2a). We find that pause sites are enriched ($p$-value $< 0.01$ using a hypergeometric test) downstream of specific secondary structure motifs, such as α-helices and β-sheets (Fig. 2b and Supplementary Fig. 2), yet are not significantly enriched at the termini of domains (Supplementary Fig. 3 and also see Supplementary Note 1). On average, pausing becomes most substantial six to eight amino acids downstream of α-helices and β-sheets, which, in the majority of cases, fall either on disordered regions of the protein or on helical residues. Such instances consistently account for $> 35–40\%$ of pause sites across different conditions (Fig. 2c and Supplementary Fig. 4). These findings strongly corroborate a growing theory that partially folded intermediate protein structures begin to immediately fold inside the ribosome exit tunnel, following polypeptide-chain synthesis. Several previous studies have shown that partially folded protein structures, such as small domains, can be detected within the exit tunnel[21–23]. More recently, Nilsson *et al.*[24] demonstrated the co-translational folding of a small zinc finger-like domain deep within the ribosome exit tunnel using arrest-peptide-mediated force measurements in conjunction with cryo-electron tomography.

Do sequence-specific motifs drive co-translational pausing to ensure proper protein folding? We find that Shine–Dalgarno (SD)-like sequences account for 20–22% of ribosome density at pause sites (Fig. 2c and see 'Identification of SD-like codons' in Methods), which is consistent with recent studies[25], and four times less frequent than what is found previous studies[20]. Of the pausing instances linked to SD-like sequences, we find that, on average, nearly half of these pausing regions also fall in the nearby vicinity (five to ten codons) of helices or sheets. The link between pausing, SD-like sequence and protein secondary structure becomes clear when comparing the average occurrence of SD-like sequences genome-wide (9%) with their occurrence directly downstream of α-helices (35%) and β-sheets (18%, Fig. 2d). Together, these sequence and structure motifs account for the majority of pause sites (60%) or nearly half of the total ribosome occupancy (Supplementary Fig. 4). These findings suggest that co-translational pausing occurs for distinct secondary structural elements and supports the potential role of sequence-specific factors to drive pausing for ensuring proper protein folding (Fig. 2e).

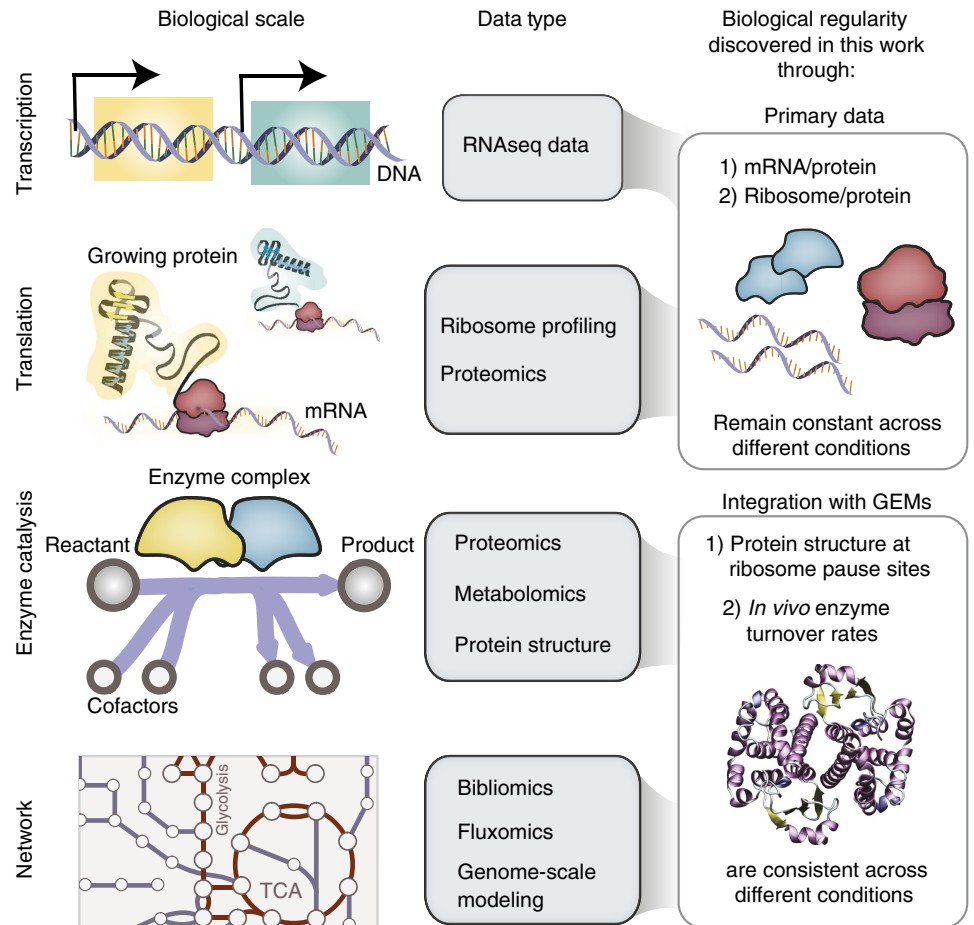

**Figure 1 | A multi-scale multi-omics framework detects significant biological regularities in *E. coli*.** Tracing the central dogma of biology (left column), we can link specific data types (middle column) to explain each of these biological processes. In this work, novel biological regularities that relate these processes are discovered through: (i) primary omics data (top box, right column) and (ii) integration with GEMs (bottom box, right column).

**Genome-wide estimation of enzyme turnover rates**. How does information flow between an individual enzyme's catalytic activity and the activity of an entire network? To evaluate the effective turnover rate of enzymes, reaction flux per enzyme can be directly computed using experimental values for both flux (the rate of reactions) and enzyme abundance[26] on a small scale (mainly for central carbon metabolism). To assess enzyme turnover on a genome scale, we computed the ratio of an enzyme's abundance (measured from proteomics data) and its corresponding flux derived from network-based analyses using the *i*OL1650-ME model (Supplementary Note 2 and Fig. 3). As the *i*OL1650-ME model directly relates enzyme synthesis and metabolic flux, we were able to develop a method, which uses the model to extrapolate the most probable flux state from proteomics. These ratios quantitatively couple experimentally derived flux estimates and protein abundances to make a quantitative connection between data types.

Estimates of enzyme turnover rates ($k_{eff}$), which represent coupling coefficients between the fluxome and the proteome (Supplementary Note 3), were analysed across four nutrient conditions, to understand the effect that carbon uptake has on metabolic enzyme turnover rates. We find that these parameters show considerable regularity in relating flux to protein abundance, which suggests that *in vivo* turnover rate for most enzymes does not strongly depend on growth in diverse batch culture settings (Supplementary Note 4). For high-flux metabolic reactions, the estimated turnover rates were consistent across all four conditions (a total of 284 turnover rate values; Fig. 4a), with high correlation between any two conditions (Fig. 4b and Supplementary Fig. 5). The computed turnover rates were averaged across experimental conditions to give the largest set of flux-per-enzyme parameters estimated computationally to date under *in vivo* conditions. It is important to note that these estimated turnover rates do not have a direct relationship with fundamental enzyme kinetic parameters obtained *in vitro* but can be viewed as an *in vivo* data-driven estimate of the enzyme turnover rate.

Although these correlations provide information about relationships between biological components and, in some cases, take on predictive value (Fig. 5a), understanding their collective influence on cell physiology is harder to decipher. This issue can be addressed using a genome-scale model that assesses cost-benefit tradeoffs from a cell-centric perspective[9,27]. Genome-scale models (*i*OL1650-ME) compute the value of cellular components relative to the function of all other cellular components. To this end, the turnover rate values provide the minimum 'capital expenditure' for protein synthesis required to achieve a unit of flux through a given reaction. Thus, as a group, the calculated turnover rates provide coupling between proteome allocation and achievement of a physiological state.

The knowledge of the biological regularities identified in this work enables the parameterization of coupling constraints used in a genome-scale model of metabolism and gene expression metabolism and gene expression (ME). A parameterized model

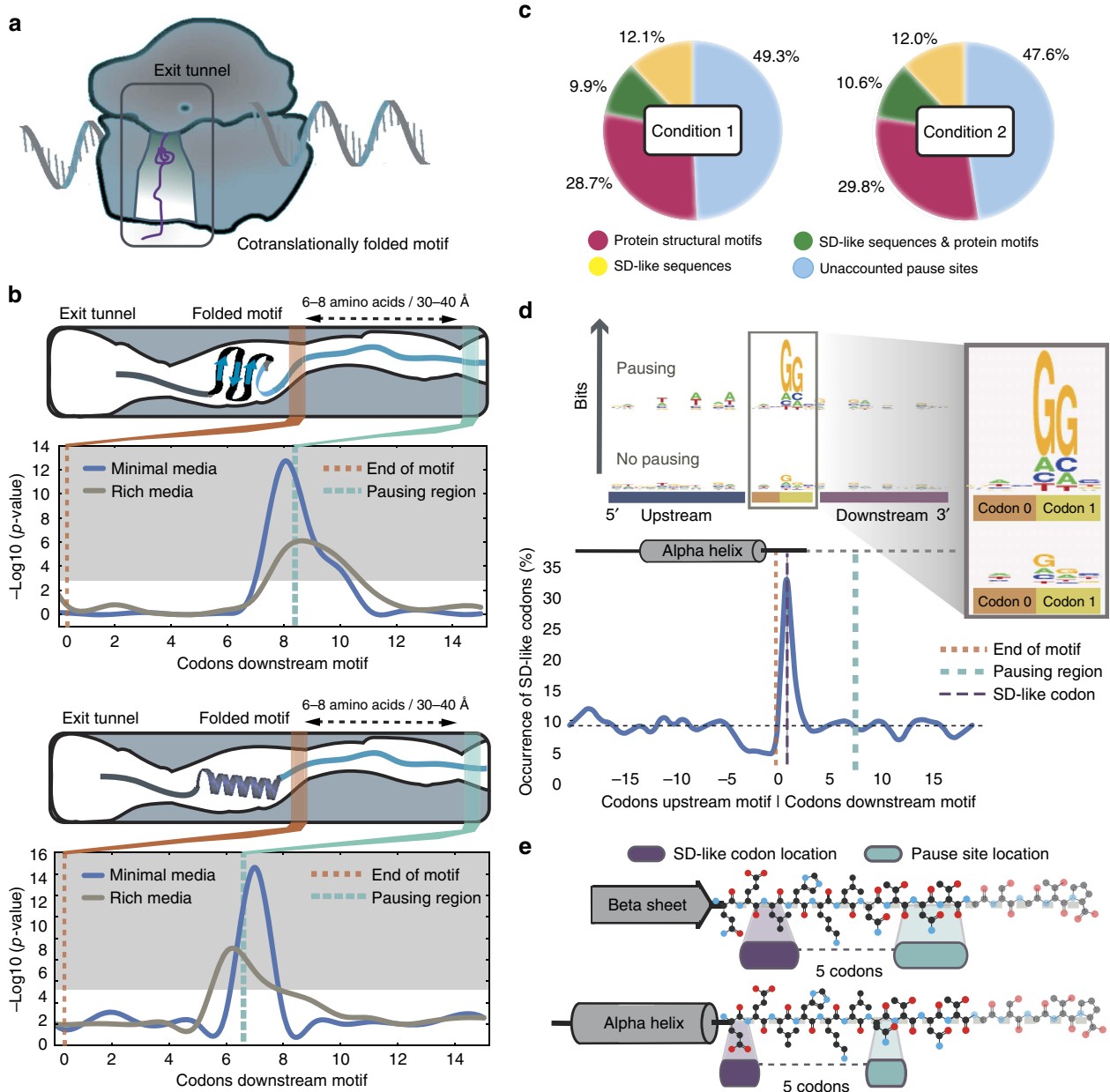

**Figure 2 | Regularities in translational pausing and structural motifs.** (**a**) Cartoon depiction of co-translational folding intermediates, such as secondary structure motifs, inside the ribosome exit tunnel. (**b**) Analysis of ribosome profiling and translational pausing in conjunction with protein structure properties in *E. coli* grown under MOPS Rich and MOPS Minimal media, taken from Li *et. al.*[11]. Pausing is enriched at positions downstream of protein secondary structures (top: β-sheets, bottom: α-helices, *p*-value < 6.67 × 10$^{-3}$). These correlations are consistent across conditions (for example, minimal and rich nutrient conditions). (**c**) Coverage of specific secondary structure elements and sequence elements that account for increased ribosome occupancy. Condition 1 refers to minimal media and condition 2 refers to rich media. (**d**) Protein structure motifs that exhibit pausing have increased propensity for SD-like sequences compared with those which do not exhibit pausing or the global background existence, 35% SD-like codons for α-helices, 18% SD-like codons for β-sheets, compared with 9% global average. (**e**) A cartoon depiction of the relationship between structure, translation and sequence.

allows for prediction of responses to environmental perturbations. We tested the predictive capacity of a model containing parameter values derived from multiple conditions described above, to compute optimal cellular composition under new environmental conditions where proteomics data was not available. In simulations with our parametrized model, we perturbed a reference growth state through the addition of nutrients to the medium: batch growth on glucose was supplemented with adenine, glycine, tryptophan or threonine. From these predicted phenotypic states, we identified enzymes that were predicted to be differentially used in the supplemented

condition (Fig. 5b,c). To validate these predictions, we collected omics data sets under these four perturbed conditions, to compare gene expression changes with the computed responses.

When validating our predictions using experimentally measured differential gene expression, we find our prediction accuracies of differential gene expression range between 56 and 100% (Table 1), and are significantly enriched for experimentally differential genes, with *p*-values ranging from 0.04 to 4 × 10$^{-6}$ using a hypergeometric test (Fig. 5b). Therefore, using the parameterized model, we are able to improve the prediction of gene regulation that accompanies changes in growth

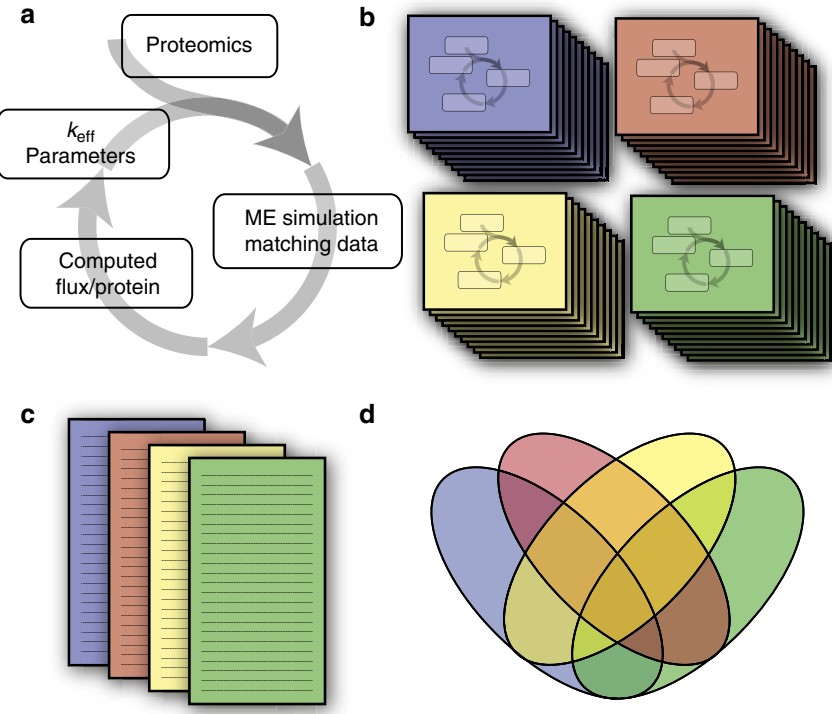

**Figure 3 | Iterative workflow for generating turnover rate values from different nutrient conditions.** (**a**) An iterative procedure uses a model to find a flux state, which most closely matches experimental proteomic data, and uses it to obtain an estimated parameter set. (**b**) The iterative workflow is run using proteomic data from four different experimental conditions. To eliminate bias from the initial parameters used in the iterative workflow, the starting parameter vectors are sampled from a uniform distribution. (**c**) For each condition, a consensus parameter set is aggregated. (**d**) Parameters are compared between conditions to obtain a universal set of condition-invariant parameters.

environment. For example, a nutrient supplementation will often cause non-intuitive shifts in what precursors the cell uses to synthesize amino acid molecules. The parameterized ME model correctly identifies the production/non-production of L-serine from supplemented L-threonine and glycine (Fig. 5c and Supplementary Note 5).

## Discussion

The unprecedented growth in the type, size and complexity of biological data sets over the past couple of decades has led to a pressing grand challenge in biology referred to as BD2K. In this study, we address this critical need through the development of advanced data integration methods to enable multi-level analysis of genomic, transcriptomic, ribosomal profiling, proteomic and fluxomic data across multiple experimental conditions. We can show that pairwise integration of primary omics data reveal unknown biological regularities that quantitatively tie key cellular processes together, and that genome-scale models enable the quantitative synchronization of disparate omics data types, leading to the discovery of additional system-based novel regularities. For example, when directly compared, RNA and protein values have long been known to be poorly correlated[16,28]. However, the integration of ribosome profiling and structural data demonstrates how the secondary structures in proteins correlate with sites of translational pausing, supporting the theory of co-translational folding and the rhythm of translation[29,30] (Supplementary Note 1). The variation in translation observed from multiple omics types sheds light on how the protein-per-RNA ratio is correlated for each gene across conditions, but poorly correlated across genes[31].

By designing algorithms to integrate omic data with genome-scale networks, the ability to predict differential gene expression

(Fig. 5c) emerges. On the most basic level, a genome-scale model of metabolism informs the user of all possible metabolic reactions and their respective stoichiometries[8]. The previous unparameterized ME model of *E. coli*[19] added an additional reconstruction of gene and protein expression networks, and their associated metabolic costs. Here, we integrate omics data directly by using a parameterization algorithm to improve estimates of expression costs. We can demonstrate the increased accuracy of the parameterized ME model compared with the previous unparameterized ME model and M-model (Table 1 and Supplementary Fig. 6).

Taken together, we have shown that fundamental contextualization of multi-omic data leads to (i) insights into underlying biological mechanisms during protein translation and (ii) predictive computations based on cellular-econometric cost-benefit ratios associated with cellular functions. Thus, both multi-omic data analysis and genome-scale models will play an important role in establishing big data analysis frameworks to explain and predict cellular physiology.

## Methods

**Ribosome profiling.** To compute the ribosome per protein ratios, *E. coli* MG1655 cells were grown in glucose ($5\,g\,l^{-1}$), pyruate (sodium pyruvate $3.3\,g\,l^{-1}$), fumarate (disodium fumarate $2.8\,g\,l^{-1}$) and acetate (sodium acetate, $3.5\,g\,l^{-1}$). Ribosome profiling data sets were generated and analysed according to the following procedure, also detailed in Latif *et al.*[13]. Chloramphenicol ($100\,\mu g\,l^{-1}$) was added 2 min before harvest, cells were harvested at mid-log ($OD_{600}\sim0.4$) by centrifugation at $5,000\,g$ for 3 min. Ice-cold lysis buffer (25 mM Tris pH 8.0, 25 mM $NH_4Cl$, 10 mM MgOAc, 0.8% Triton X-100, $100\,U\,ml^{-1}$ RNase-free DNase I, $0.3\,U\,\mu l^{-1}$ Superase-In, 1.55 mM chloramphenicol and $17\,\mu M$ 5'-guanylyl imidodiphosphate) was added and the cells were resuspended quickly, followed by flash freezing in liquid nitrogen. Repeated freeze–thaw cycles were used to lyse cells followed by addition of sodium deoxycholate to a concentration of 0.3%. Lysate was then clarified by centrifugation. RNA (25 AU) was digested using 6,000 U of

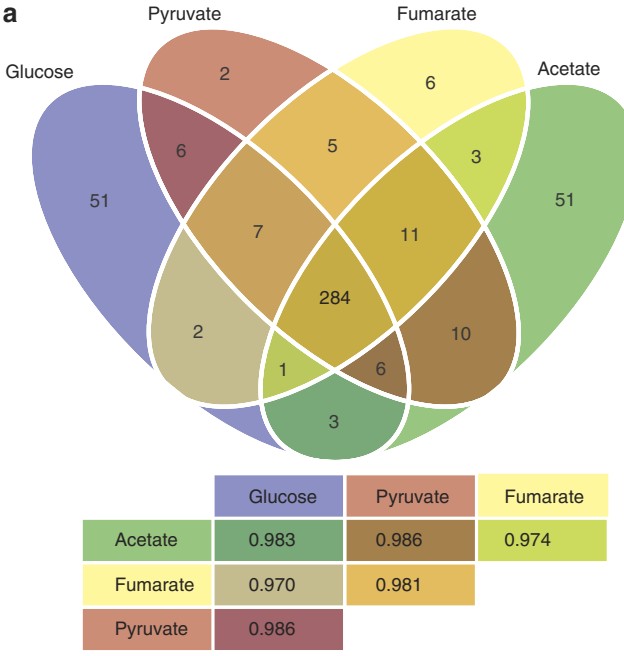

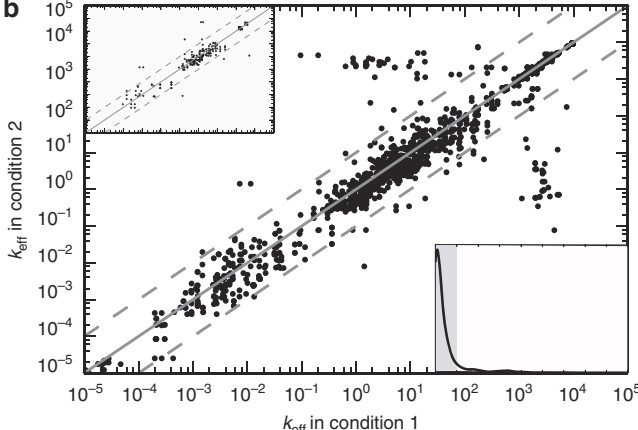

| | Glucose | Pyruvate | Fumarate |
|---|---|---|---|
| Acetate | 0.983 | 0.986 | 0.974 |
| Fumarate | 0.970 | 0.981 | |
| Pyruvate | 0.986 | | |

**Figure 4 | Effective enzyme turnover rates ($k_{eff}$) as regularities emerging from coupling quantitative in vivo proteomic data with genome scale modelling.** (a) Venn diagram of calculated turnover rates shows all four conditions share 90% of the same estimates (Pearson's correlations below). (b) Pairwise comparisons across four conditions for calculated turnover rate parameters demonstrate 94% are within one order of magnitude. The upper inset show the parameter estimation for the 10% most variable components of the proteome between the four conditions examined. The lower inset show a histogram of the distances of every point from the diagonal line. The grey box contains the 94% of the values that deviate from one another within an order of magnitude. A more detailed version is found in Supplementary Fig. 6.

MNase for 2 h at 25 °C and quenched by adding 2.5 µl of 500 mM EGTA. Monosomes were recovered using Sephacryl S400 Microspin columns, followed by removal of ribosomal RNA using a Ribo-Zero-rRNA Removal Kit (Epicentre). The 3′-ends of the RNA fragments were desphosphorylated using T4 Polynucleotide Kinase for 30 min at 37 °C. The NEBNext Small RNA Library Prep Set for Illumina protocol was carried out till the 5′-adapter ligation step. Transfer RNAs were removed using hybridization to custom DNA oligo probes, followed by RNASE H treatment. The NEBNext protocol was then completed and sequencing was carried out on a Illumina MiSeq.

As the MiSeq did not provide sufficient read depth for the in-house ribosome profiling data sets to confidently determine pause locations, ribosome profiling data was obtained from Li et. al.[11] for MOPS Rich and glucose MOPS minimal

media (GEO Accession: GSE53767). Ribosome densities were derived using a similar protocol to Li et. al.[11]. Adapters were trimmed using cutadapt[32] version 1.8. Reads were mapped using bowtie[33] version 1.0.0 to E. coli MG1655 (NC000913), allowing for a maximum of one mismatch. Reads mapping to tRNA, ribosomal RNA and other non-coding RNA locations were discarded. There has been considerable discourse about the location of the A- and P-site relative to the ends of the reads, and not to bias our analysis based on the location we chose to assign reads to the 3′-end, which has been shown to be better conserved and aligned in prokaryotic ribosome profiling data sets[34,35]. Ribosome density across each gene was then dropoff corrected by fitting to an exponential function as was done in Li et. al.[11].

**Genome-wide secondary structure annotations.** The GEM-PRO reconstruction for E. coli iJO1366 (refs 17,36) was used to provide structure-based annotations for the most representative protein structures found in the publicly available PDB database[37]. Protein data bank (PDB) files were parsed using STRIDE[38] and Biopython[39], to determine the location of secondary and tertiary structural elements on a codon-specific basis. This resulted in high-confidence secondary and tertiary structure annotations for 623 non-transport (or membrane-bound) genes in E. coli.

**Tertiary structure and protein domain annotations.** Starting from the protein structures linked to metabolic genes in the GEM-PRO model, we annotated tertiary domains for each protein using the SCOP knowledgebase[40] and FATCAT[41] alignment tools. As a result of this analysis, the fraction of the protein aligning to an annotated tertiary domain was recorded and stored as an additional type in the GEM-PRO reconstruction. The starting and ending amino acid of every domain within a protein were quality controlled and checked by aligning the PDB sequence with the amino acid sequence (FASTA) from E. coli MG1655 to fix offsets between the PDB residue numbering scheme and the actual amino acid sequence numbers. Hypergeometric enrichment testing was performed, to determine codons upstream and/or downstream from the start and end of any tertiary domain annotation that is enriched for pause sites.

**Identification of SD-like codons.** Similar to those defined in Li et. al.[20], we considered the following SD-like codons: 5′-'AGG','GGA','GAG','GGG','GGT', 'GTG'-3′. Nucleotide sequences from E. coli MG1655 (Genbank accession: NC000913 (ref. 42)), were read in-frame to identify SD-like codon positions. Hypergeometric enrichment testing was used to determine downstream codons enriched for pause sites.

**Ribosome density and pause site accounting.** The ribosome density (Supplementary Figs 2 and 3) at each codon was summed across all three nucleotides and divided by the mean of the gene, to get the normalized density. In an effort to increase the signal in an inherently noisy data type, pause sites were defined as codons, which had a normalized density of over 5, instead of on a per nucleotide basis as was done in Li et. al.[20].

Hypergeometric enrichment testing was used to determine enrichment for pause sites at codon positions downstream from the ends of pause sites. P-values were calculated based on the formula for the survival function (1 − cumulative distribution function) shown below:

$$p\text{-}value(x) = 1 - \sum_{i=0}^{x-1} \frac{\binom{m}{i}\binom{N-m}{k-i}}{\binom{N}{k}}$$

where N refers to the total number of codons in the genes tested, m refers to the number of secondary structures, k refers to the total number of pause sites and x refers to the number of pause sites that fell on a specific codon position downstream of the secondary structure we are testing.

Codons downstream from α-helices, β-sheets and turn secondary structural elements, as well as SD-like sequences, were considered to be significantly enriched for pausing, if the hypergeometric enrichment tests indicated that the p-values <0.01. To calculate the number of pause sites accounted for by sequence and/or structural elements, pause sites that did not align with secondary structure or SD-like sequences were labelled 'unaccounted'. The same procedure was used for determining the proportion of pause sites accounted for by structural features and SD-like sequence features.

**Computational method for predicting $k_{eff}$ parameters.** As noted above, ME simulations require several parameters, one of which is the effective catalytic rate of enzymes $k_{eff}$, which in turn affects the proteomic and ribosomal cost of running each reaction. Although solvent accessible surface area (SASA) as a first approximation results in a correct overall prediction of 80% of the cell proteome by mass, improving these parameters can greatly affect the predictive power of the model for specific genes. We make use of the most extensive quantitative proteomic data in E. coli to date, which accounts for 55% of all open reading frames and 95% of the proteome[14]. Because of the difficulty in simultaneously predicting effective catalytic rates and reaction flux values, we developed an iterative workflow for

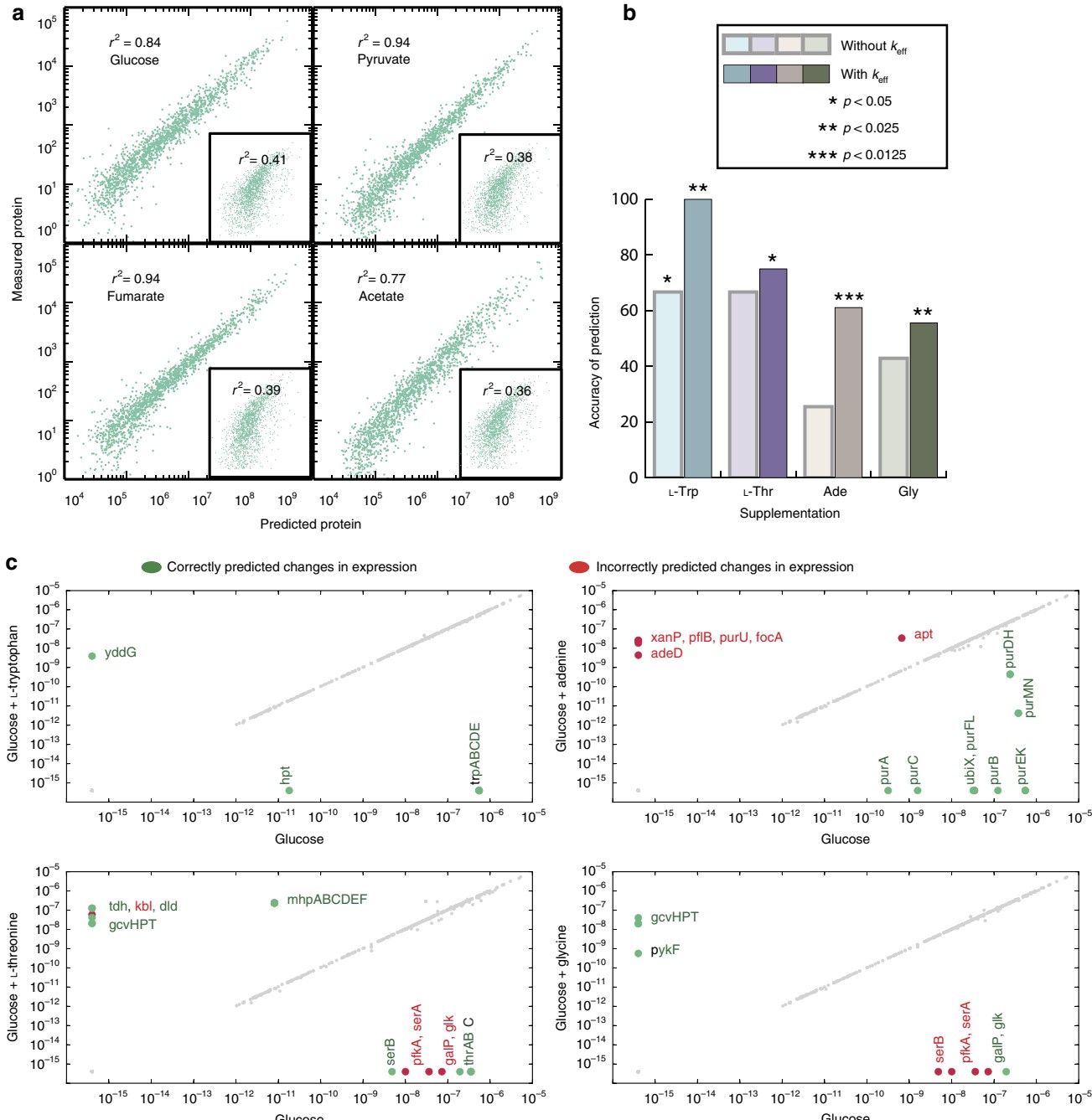

**Figure 5 | Predicting the results of perturbation from a parameterized homeostatic state.** (**a**) Using a cross-validation approach, protein abundance is predicted by mRNA levels using information ($\rho_{PM}$) obtained from other conditions ($r^2 > 0.75$). Condition-specific mRNA and protein levels show little correlation (inset). (**b**) Accuracy of predicting differential expression is significantly enhanced using $k_{eff}$ parameters. (**c**) Predictions of differential gene expression using enhanced $k_{eff}$ parameters after media supplementation. The predictions were validated using expression profiling and gave predictive accuracies range between 56 and 100%. In all cases, predictions of differentially expressed genes are significantly enriched for those which have been experimentally confirmed ($p < 0.05$ using a hypergeometric distribution).

updating the model parameters. This workflow is described below and each section corresponds to a panel in Fig. 3.

**Part A iterative simulation procedure.** These ME simulations had the overall goal of minimizing the difference between the simulated proteome and the measured proteome for each experimental condition. The growth rate $\mu$ was set to the experimentally determined values (Supplementary Data 1). To improve solution times with the SoPlex linear programming solver[43] with our formulation,

we collapsed linear pathways into single reactions using cobrapy[44], which were detected by identifying metabolites present in exactly two reactions.

To reconcile experimentally measured protein concentrations with the simulations, we want to solve a linear program, which will minimize the Manhattan distance between the expressed and measured protein production at the fixed growth rate. The Manhattan distance was used instead of the Euclidian distance, because it can be computed in the context of a linear program, whereas a Euclidian distance minimization requires a quadratic objective, which the SoPlex solver can not handle. To construct this problem, we added the following additional constraints to the ME linear program in terms of the corresponding measured

**Table 1 | Predicted expression changes confirmed experimentally.**

| | $k_{eff}$ parameters from set A | | | | $k_{eff}$ parameters from set A + B | | | |
|---|---|---|---|---|---|---|---|---|
| | Correct | Incorrect | Accuracy | *p* | Correct | Incorrect | Accuracy | *p* |
| ʟ-Tryptophan | 6 (0) | 3 | 66.7% | 0.031 | 7 (0) | 0 | 100.0% | 0.011 |
| ʟ-Threonine | 30 (5) | 15 | 66.7% | 0.119 | 15 (5) | 5 | 75.0% | 0.044 |
| Adenine | 12 (4) | 35 | 25.5% | 0.663 | 11 (4) | 7 | 61.1% | $4 \times 10^{-6}$ |
| Glycine | 15 (0) | 20 | 42.9% | 0.391 | 5 (0) | 4 | 55.6% | 0.024 |

The ME model predicted differential expression for the following nutrient supplementations to growth of *E. coli* on M9 minimal media with ᴅ-glucose. Two separate sets of $k_{eff}$ were used, one using only the 28 parameters in set A and the other also adding in the 284 modelling-derived parameters in set B. The predictions were evaluated using the set of differentially expressed genes determined from mRNA sequencing for each nutrient supplementation. Two small modifications were made to the model for adenine, which initially had an accuracy of only 26.2% due to the genes used for adenine degradation in the model, which are not expressed and may not be functional (for details, see Supplementary Fig. 7). The numbers provided in the correct column are the number of genes, which are predicted to be differentially expressed and also are differentially expressed in the same direction in the RNA sequencing data. The number in parenthesis refers to the number of correctly predicted differentially expressed genes which are still expressed under both conditions but varied in their predicted quantitative values. The incorrect column contains the number of genes predicted to be differentially expressed which were not in the model. These numbers are used to predict the percent accuracy and the *p*-value for a hypergeometric enrichment of differentially expressed genes in the predicted set.

protein amounts $y_i$ for each predicted gene translation flux variable $x_i$ (unmeasured proteins had no applied constraints).

$$x_i \geq 0$$

$$s_{+i} \geq 0$$

$$s_{-i} \geq 0$$

$$x_i + s_{+i} \geq y_i$$

$$x_i - s_{-i} \leq y_i$$

This allowed us to minimize the error term $\sum_i (s_{+i} + s_{-i})$, which is equivalent to minimizing $\sum_i |x_i - y_i|$ and gives the closest flux state to the experimental data, while satisfying the ME constraints when solving the linear program. If these parameters resulted in an infeasible model, which could not simulate growth at $\mu$, the simulation was halted. Otherwise, using the predicted fluxes predicted by this simulation and the experimentally measured proteomics data, we calculated the $k_{eff}$ for each reaction enzyme, which we used in the next iteration of this workflow. This simulation loop was run a total of three times, to allow the loops to converge to a set of $k_{eff}$ values.

**Part B sampling and simulation.** The iterative simulation procedure described above might give a flux state, which is dependent on the original set of $k_{eff}$ parameters used in the first round. Therefore, the $k_{eff}$ were randomly initialized within two orders of magnitude of the value computed from SASA. The iterative sampling procedure was repeated repeated 300 times for each experimental condition and each time a new random $k_{eff}$ parameter set was generated in the manner described.

**Part C result aggregation and filtering.** For each experimental condition, the loop was started 300 times. However, as some parameter sets were unable to simulate growth at $\mu$, some subset of those simulations failed before reaching three iterations. It was run successfully through 3 loops 148 times for glucose, 186 times for pyruvate, 97 times for fumarate and 83 times for acetate. Between these successful runs, there was a slight variation in reactions used because of the different starting $k_{eff}$ parameters. Therefore, only reactions that were active for 90% of the successful runs were considered. These parameters were averaged to give a consensus set of $k_{eff}$ parameters for each condition.

**Part D cross-condition parameter comparison.** The intersection of these $k_{eff}$ parameters under each condition was determined between all four conditions (Fig. 4a). The 284 parameters for reaction/catalyst pairs, which were in all conditions, were averaged to get a consensus set of $k_{eff}$ parameters, which were used for ME computations. In addition, the pairs in common between each of the conditions were compared to give Pearson's correlations, shown in the table in Fig. 4a.

**Predicting differential gene expression with *iOL1650-ME*.** Simulations were performed using the same procedure as in the *iOL1650-ME* manuscript[19] for batch growth on ᴅ-glucose with both $k_{eff}$ parameter sets A and A + B. For each supplementation simulation, the uptake reaction for that particular metabolite was set to be unbounded. In the case of the adenine supplementation, the reactions HXAND, XAND and URIC were blocked (Supplementary Fig. 7). Genes that changed by more than a factor of 16 (a log₂ change of more than 4) were predicted to be differentially expressed. This gives a stringent criterion, which identifies genes that are predicted to change by a significant-enough magnitude to manifest experimentally (by comparison, a log₂ change of 2 is often used with microrray gene expression data to filter out changes in expression, which, although statistically significant, are not of a high-enough magnitude to really be considered relevant). Predictions of gene differential expression were considered correct if cufflinks obtained a false discovery rate of <0.05 for that gene in the mRNA sequencing data

and the gene expression changed in the same direction (either both increase or both decrease) in both the predictions and mRNA sequencing data. Hypergeometric enrichment *p*-values were calculated using the scipy statistics package using the survival function + ½ × probability mass function of the distribution.

**Data availability.** RNA sequencing data generated in this study is available from the NCBI Gene Expression Omnibus (GEO) under accession numbers GSE59759 and GSE59760. Measured growth rates are available in Supplementary Data 1 and estimated $k_{eff}$ parameters in units of per second are available as Supplementary Data 2. All other relevant data are available from the authors upon request.

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

## Acknowledgements

This work was funded from a generous gift from the Novo Nordisk Foundation to the Center for Biosustainability (grant NNF16CC0021858). It was also supported by DE-FOA-000014 from the U.S. Department of Energy, NIH R01 GM057089 from the National Institutes of Health and by the Swiss National Science Foundation (grant p2elp2_148961 to E.B). This research used resources of the National Energy Research Scientific Computing Center, which is supported by the Office of Science of the US Department of Energy under Contract Number DE-AC02-05CH11231. We gratefully acknowledge Dr Mahmoud Al-Bassam and Dr Jinwoo Kim for scientific discussions on ribosome profiling.

## Author contributions

Conceptualization, A.E., E.B., J.T. and B.O.P. Methodology (ME modelling: A.E., J.L., E.J.O. and A.M.F.; ribosome profiling analysis: E.B., J.T.). Investigation, J.T., A.E. and E.B. Writing original draft, E.B., A.E. and B.O.P. Writing, review and editing, E.B., A.E., J.T., J.L., A.M.F. and B.O.P. Discussion, all authors. Funding acquisition, resources and supervision, B.O.P.

## Additional information

**Competing financial interests:** The authors declare no competing financial interests.

