## [Peer review file · Nature Communications]

Reviewers' Comments:

Reviewer #1 (Remarks to the Author)

The article deals with one of the biggest challenges of modern biology, namely the question of how to best utilize and combine data collected from a variety of sources and biological systems in order to magnify our understanding of biological function as a whole. The authors try to address this question by identifying biological regularities, ie relationships among diverse data types that remain constant at a variety of conditions, and by using a ME model of E.coli to make predictions for changes in gene expression for different growing conditions.

The article is written very well; it is well-presented and easily understandable. The results that are presented therein are robust and convincing and constitute incremental but significant progress for the field. I do not have any major revisions to suggest to the authors, except perhaps the first comment regarding accessibility of the model(s) used. Otherwise, I believe the manuscript should be acceptable for publication by Nature Communications.

Discretionary revisions:

1) The authors should make the model or models used for the study easily accessible, by providing a way to download or by inclusion in a public database (eg biomodels.net). It becomes clear after careful study of the supp. materials that the model used was primarily iOL1650-ME model, but this information is completely skipped in the main body of the paper, and it would benefit the casual or novice reader if a direct link was provided, preferably with all the necessary modifications for the model to reproduce the results in the paper.

2) Even though the paper is generally well-written some improvements could be made for clarity purposes. For example, it is not completely clear what organism is being used for the study, because the name "E.coli" is used only once in the entire paper, and that is in the abstract. Similarly, the nature of the model used (an ME model) is not mentioned anywhere in the paper, but it is explained well in the supp. materials. It is also not completely clear how the various pieces (ie the hidden regularities) presented come together, because this is summarized in a very short last paragraph. I understand all this is most likely a result of the limited space available to the authors (especially since the supp. materials do not suffer from the same problems) and it may not be possible to make significant improvements.

3) Minor corrections:

- Top of page 5: "on a genome scale" (no hyphen needed)

- "Keffs" is used in a few instances in the manuscript and it can be confusing; "turnover rates" would be much better instead

Reviewer #2 (Remarks to the Author)

In this study, Ebrahim et al. used multiple omic data sets from E. coli to interrogate growth condition-associated variants and invariants. The main results were divided into translational analysis and metabolomic analysis. Failed to find any specific biological response, the authors reported invariants or regularities across the different growth conditions. Although it is of fundamental importance by integrating different types of big data, this manuscript does not seem to bring up any novel findings. The reported correlation between protein structure, SD-like sequence and ribosome pause are not new at all. Some conclusions suffer from false interpretation. For instance, the pausing site is located about 8 codons downstream the structural domain. However, the ribosome peptide tunnel accommodates at least 30 amino acids. Therefore, it is contradictory to the main conclusion that this kind of ribosome pausing helps co-translational folding that occurs outside of the tunnel. As a result, the entire study does not meet the initial goal stated in the introduction. Besides some randomly stacked data, there is limited conceptual

advance in our understanding of the mechanistic connection between genetic flow and metabolic flux.

Specific concern:

1. Page 3. The authors described three correlations: the correlation of mRNA to protein (<0.4), correlation of protein/mRNA under different conditions, and correlation of ribosomes per proteins under different conditions (>0.7). However, these correlations have different biological meanings and each pair-wise analysis is different (the first vs. the second, or the first vs. the third). It is difficult to understand how these irrelevant correlations lead to "synchronization" of disparate data sets.

2. page 4. The authors performed hypergeometric test to find the lower p-value regions. Because there are many comparisons, it would be better do multiple corrections on p-values to reduce the false positive rates

3. The SD sequence analysis is kind of questionable because if the authors used the data generated by Li et al (2014). A recent study reported that the internal SD-like sequence only accounts for a small fraction of pause sites (<http://www.cell.com/cell-reports/fulltext/S2211-1247%2815%290152-9-6>).

The Cell Reports paper also analyzed many other bacteria ribo-seq datasets and found very weak or even no correlation between SD-like sequence and pause site. Therefore, the authors have to analyze more datasets derived from various protocol and biological conditions to make their conclusion about the correlation between SD-like sequence, alpha-helices and beta-sheets more robust and convincing.

4. Page 6, Using proteomics data (protein level) to predict differential gene expression (mRNA level) seems to be really baffling. The computational model should be designed to predict data which is more difficult to obtain or measure from data which is easier to generate. Given gene expression data (RNA-Seq) are much easier to produce with high-throughput sequencing, it is hard to imagine the real value of this predictive model.

5. The manuscript contains many supplementary results. But surprisingly there is no description at all in the main text for supplementary fig 6 - fig 11.

Reviewers' comments:

Reviewer #1 (expert in genome-scale models, big data integration and systems biology) (Remarks to the Author):

The article deals with one of the biggest challenges of modern biology, namely the question of how to best utilize and combine data collected from a variety of sources and biological systems in order to magnify our understanding of biological function as a whole. The authors try to address this question by identifying biological regularities, ie relationships among diverse data types that remain constant at a variety of conditions, and by using a ME model of *E.coli* to make predictions for changes in gene expression for different growing conditions.

The article is written very well; it is well-presented and easily understandable. The results that are presented therein are robust and convincing and constitute incremental but significant progress for the field. I do not have any major revisions to suggest to the authors, except perhaps the first comment regarding accessibility of the model(s) used. Otherwise, I believe the manuscript should be acceptable for publication by Nature Communications.

Response:

We thank the reviewer for their kind praise.

Discretionary revisions:

1) The authors should make the model or models used for the study easily accessible, by providing a way to download or by inclusion in a public database (eg biomodels.net). It becomes clear after careful study of the supp. materials that the model used was primarily iOL1650-ME model, but this information is completely skipped in the main body of the paper, and it would benefit the casual or novice reader if a direct link was provided, preferably with all the necessary modifications for the model to reproduce the results in the paper.

Response:

We acknowledge the reviewer's interest in the accessibility of the *E. coli* ME model. Unfortunately, the iOL1650-ME model used in this study can not be included in a database like biomodels.net because the SBML standardization has not yet occurred, as ME models are still so new and quite complicated. However, we are happy to report that our lab has been working on reformulating ME models in a manner which will make it much easier for the community to download, install and use. This has been a major (separate) effort, but we are happy to say it is near completion and we are submitting a manuscript describing this soon. Additionally, because creation of standards is a community process, we have begun discussing with some of the SBML editors ideas for how the reformulated model can be standardized, and we hope that after our other manuscript describing the reformulation is published, the community can use our reformulation as a template and converge on a solution to this issue. We thank the reviewer again for their keen interest in the model.

Additionally, we thank the reviewer for pointing out that the model used would have been unclear to a casual reader of the main text. We have added some details to the main text, as highlighted in blue font.

2) Even though the paper is generally well-written some improvements could be made for clarity purposes. For example, it is not completely clear what organism is being used for the study, because the name "*E.coli*" is used only once in the entire paper, and that is in the abstract. Similarly, the nature of the model used (an ME model) is not mentioned anywhere in the paper, but it is explained well in the supp. materials. It is also not completely clear how the various pieces (ie the hidden regularities) presented come together, because

this is summarized in a very short last paragraph. I understand all this is most likely a result of the limited space available to the authors (especially since the supp. materials do not suffer from the same problems) and it may not be possible to make significant improvements.

Response:

We strive to make the manuscript as clear as possible and welcome the reviewer's suggestions to improve the clarity in the manuscript in multiple sections. The organism used in the study has been mentioned now (highlighted throughout the text in blue colored font). In addition, we have added to the main text some details on the modeling, and have also added references in the main text to the full detailed description in the supplementary methods. We have also extended the discussion of how the various pieces of this study come together. We hope that the reviewer finds the revised text more clear, cohesive and improved.

3) Minor corrections:

- Top of page 5: "on a genome scale" (no hyphen needed)

Response:

We have made the corrections in the text.

- "Keffs" is used in a few instances in the manuscript and it can be confusing; "turnover rates" would be much better instead

Response:

We have made the corrections in the text.

Reviewer #2 (expert in translational control in gene expression, translational pausing and ribosome profiling) (Remarks to the Author):

In this study, Ebrahim et al. used multiple omic data sets from *E. coli* to interrogate growth condition-associated variants and invariants. The main results were divided into translational analysis and metabolomic analysis. Failed to find any specific biological response, the authors reported invariants or regularities across the different growth conditions. Although it is of fundamental importance by integrating different types of big data, this manuscript does not seem to bring up any novel findings. The reported correlation between protein structure, SD-like sequence and ribosome pause are not new at all. Some conclusions suffer from false interpretation. For instance, the pausing site is located about 8 codons downstream the structural domain. However, the ribosome peptide tunnel accommodates at least 30 amino acids. Therefore, it is contradictory to the main conclusion that this kind of ribosome pausing helps co-translational folding that occurs outside of the tunnel. As a result, the entire study does not meet the initial goal stated in the introduction. Besides some randomly stacked data, there is limited conceptual advance in our understanding of the mechanistic connection between genetic flow and metabolic flux.

Response:

We thank the reviewer for pointing out how the original text could lead to misunderstanding and misinterpretation. We have revised the text to stress several points that we believe may have been missed or unclear in the original version. Stated briefly, our study is in line with several recently published manuscripts that support the idea of co-translational folding of protein structural motifs inside the ribosome exit tunnel. Here, through ribosome profiling, we complement previous studies and support the concept that the ribosome itself can provide a sheltered folding environment for smaller protein motifs (e.g., alpha helices and beta sheets) whereas chaperones may provide such an environment to larger proteins. Several studies have shown that partially folded intermediate structures can be detected already within the ribosome exit tunnel (Mingarro et al. 2000; Bhushan et al. 2000; Tu et al. 2014). More recently, Nilsson *et al.*

demonstrated that co-translational folding of small protein structural motifs (e.g. zinc finger domains) fold deep within the ribosome exit tunnel by using arrest-peptide mediated force measurements and cryo-EM (Cell Reports (2015)). These studies, together with our contribution, suggest that the co-translational folding of structural intermediates, such as alpha helices, beta sheets and smaller protein domains, is likely to begin immediately after polypeptide-chain synthesis at the ribosomal peptidyl transferase center. Our data, which indicates significant pausing 6 to 8 codons *downstream* the secondary structural motif strongly corroborates with this theory.

We believe that our findings are novel because this is the first example of showing enriched pausing at specific protein structural motifs. Previously, the only reported correlations, to the best of our knowledge, between pausing and secondary structure were very low (correlation ~ 0.1), carried out for only a few proteins and documented solely in the Supplementary Information of Li *et al.* 2012. While the link between SD-like sequences and pausing has been established (Li *et al.* 2012) and reassessed recently (Mohammed *et al.* 2016), we are mainly interested in demonstrating that a small percentage of the pausing that occurs downstream of structural motifs can be accounted for by SD-like sequence motifs. We firmly believe there are possibly multiple mechanisms by which the cell induces pausing to ensure proper folding of protein intermediates. We have revised the text and Figure 2 to reflect these points.

Specific concern:

1. Page 3. The authors described three correlations: the correlation of mRNA to protein (<0.4), correlation of protein/mRNA under different conditions, and correlation of ribosomes per proteins under different conditions (>0.7). However, these correlations have different biological meanings and each pair-wise analysis is different (the first vs. the second, or the first vs. the third). It is difficult to understand how these irrelevant correlations lead to "synchronization" of disparate data sets.

Response:

We agree that this was confusing and have changed the to address the reviewer's concerns. To explain briefly, we correlate ratios between different environmental conditions (rather than pairwise correlations of the three correlations). Though this may seem like a subtle difference, the fact that we find such good correlations *across* conditions tells us that the biological ratios (number of protein molecules to the number of RNA molecules) is constrained and relatively condition invariant. This finding addresses one of the key issues in the omic sciences (transcriptomics and proteomics) in our ability to reconcile measurements from disparate data types. Synchronization, or reconciliation, of the data, in this case, refers to finding patterns or consistencies that exist across conditions (what we term a "biological regularity"). Until now, little to no reconciliation has been found when attempting to directly correlate mRNA values to protein values ($r^2 < 0.4$). The low correlation of these values suggest that these quantities vary widely across genes and are not invariant. Yet, by taking the ratio of protein/mRNA or ribosomes/protein on a per-gene basis, we find that these values are relatively invariant in *E. coli* over different nutrient conditions. In this way, we can synchronize two disparate data types: transcriptomics and proteomics, through this common link (or ratio).

2. page 4. The authors performed hypergeometric test to find the lower p-value regions. Because there are many comparisons, it would be better do multiple corrections on p-values to reduce the false positive rates

Response:

We thank the reviewer for pointing this out, and have taken their suggestion by correcting the figures and the numbers reported in the main text. We statistically assess 15 codon positions downstream of alpha helix and beta sheet secondary structures, using the Bonferroni correction, and find that the minimum p-value for significance is now 6.67×10^{-3} . As the pause-site enriched regions show enrichments with p-values between

10^{-5} for turns and coils and 10^{-14} for alpha helices, we believe that the results remain robust to FDR corrections. The figures have been updated in the paper and supplementary text. There are now lines drawn to indicate the significance cut-offs using the corrected values (see Figures 2(b)).

3. The SD sequence analysis is kind of questionable because if the authors used the data generated by Li et al (2014). A recent study reported that the internal SD-like sequence only accounts for a small fraction of pause sites (<http://www.cell.com/cell-reports/fulltext/S2211-1247%2815%2901529-6>). The Cell Reports paper also analyzed many other bacteria ribo-seq datasets and found very weak or even no correlation between SD-like sequence and pause site. Therefore, the authors have to analyze more datasets derived from various protocol and biological conditions to make their conclusion about the correlation between SD-like sequence, alpha-helices and beta-sheets more robust and convincing.

Response:

We agree that our analysis would be strengthened by considering the additional datasets provided in Mohammed *et al.* 2016 and we now provide an extended analysis on multiple additional datasets (GEO accession: GSE72899). Through this analysis, we have found a consistently strong link between ribosome pausing and secondary structure motifs (namely, alpha helix and beta sheet structures). In each of the datasets, we find consistent pausing signature at 6 to 8 codons downstream of alpha helices or beta sheet motifs. Despite differences in protocols, environments, etc., the correlations between protein structure and pausing that we initially observed from the original dataset (Li et al. 2014) are upheld. Thanks to the reviewer's suggestion to carry out the same analysis on these additional datasets, these findings are further strengthened. We now include additional figures in Supplementary information (see Supplementary Figure 5) that demonstrate these correlations.

For the SD-like sequence correlations, we would like to clarify that, although we observed a certain degree of overlap between pausing and SD-like sequences, we believe that SD-like sequences are not likely the main mechanism employed by the cell to achieve co-translational pausing. To better stress this point, we have added text in the manuscript, alongside an additional figure (see Figure 2(c) and Supplementary Figure 4) and have referenced our findings in light of the recent Mohammed et al. 2016 paper. Here, we demonstrate that, of all pause sites, less than 30% can be accounted for by SD-like sequence motifs. While this is not in the scope of this paper, future work is likely to focus on understanding what other mechanisms the cell employs to ensure pausing at these particular sites along a transcript to ensure proper folding takes place.

4. Page 6, Using proteomics data (protein level) to predict differential gene expression (mRNA level) seems to be really baffling. The computational model should be designed to predict data which is more difficult to obtain or measure from data which is easier to generate. Given gene expression data (RNA-Seq) are much easier to produce with high-throughput sequencing, it is hard to imagine the real value of this predictive model.

Response:

We fully agree that proteomics data is much harder to obtain than gene expression data, and merely taking proteomics data and using it to predict gene expression under the same conditions would be a contrived use of a model. To this end, we believe there was a misunderstanding of what the value and use of the model is in this contribution. Briefly, we have focused on parameterizing a computational model under a few conditions using proteomics, and fluxomics. This parameterization enables the model to formulate higher accuracy predictions (compared to the status quo) in brand new growth environments, (we have shown with simulations of cells grown in glucose minimal media with four additional nutrient supplementations). We hope the reviewer agrees that being able to accurately predict protein expression in new conditions without the use of proteomics data is both challenging and important.

We have made changes in the main text to better stress this point and improve overall clarity of expressing the value of the parameterized model.

5. The manuscript contains many supplementary results. But surprisingly there is no description at all in the main text for supplementary fig 6 - fig 11.

Response:

We have now removed all supplementary figures that are not discussed. The ones that remain are discussed in Supplementary text or the main text.

Reviewers' Comments:

Reviewer #1 (Remarks to the Author)

The authors fully addressed my previous critiques.

Reviewer #2 (Remarks to the Author)

In this revised manuscript, Ebrahim et al have made improvements and addressed most of the concerns. Although the connection between translation-omics and metabolite flux is still vague, the overall concept is of significance. I now support acceptance of this manuscript in Nature Communications. My only concern is about Figure 4c. Given the importance of prediction by computational modeling, it could be better to give a more detailed description of Figure 4c in the main text. The current version is too brief to fully understand its significance.

Reviewer 1

The authors fully addressed my previous critiques.

We thank the reviewer for the gracious helpful feedback.

Reviewer 2

In this revised manuscript, Ebrahim et al have made improvements and addressed most of the concerns. Although the connection between translation-omics and metabolite flux is still vague, the overall concept is of significance. I now support acceptance of this manuscript in Nature Communications. My only concern is about Figure 4c. Given the importance of prediction by computational modeling, it could be better to give a more detailed description of Figure 4c in the main text. The current version is too brief to fully understand its significance.

We thank the reviewer for pointing out this shortcoming in the manuscript to us, and apologize deeply for the previously incomplete description. As suggested by the reviewer, we have described the computational results corresponding to figure 5c (formerly figure 4c) more fully both in the caption and in the main text to clearly state their impact.